# The Role of Protein Misfolding and Tau Oligomers (TauOs) in Alzheimer’s Disease (AD)

**DOI:** 10.3390/ijms20194661

**Published:** 2019-09-20

**Authors:** Barbara Mroczko, Magdalena Groblewska, Ala Litman-Zawadzka

**Affiliations:** 1Department of Neurodegeneration Diagnostics, Medical University of Białystok, 15-269 Białystok, Poland; ala.litman-zawadzka@umb.edu.pl; 2Department of Biochemical Diagnostics, University Hospital of Białystok, 15-269 Białystok, Poland; magdalena.groblewska@umb.edu.pl

**Keywords:** Alzheimer’s disease, tau oligomers, neurodegeneration, protein misfolding, tauopathy, CSF, plasma, serum

## Abstract

Although the causative role of the accumulation of amyloid β 1–42 (Aβ42) deposits in the pathogenesis of Alzheimer’s disease (AD) has been under debate for many years, it is supposed that the toxicity soluble oligomers of Tau protein (TauOs) might be also the pathogenic factor acting on the initial stages of this disease. Therefore, we performed a thorough search for literature pertaining to our investigation via the MEDLINE/PubMed database. It was shown that soluble TauOs, especially granular forms, may be the most toxic form of this protein. Hyperphosphorylated TauOs can reduce the number of synapses by missorting into axonal compartments of neurons other than axon. Furthermore, soluble TauOs may be also responsible for seeding Tau pathology within AD brains, with probable link to AβOs toxicity. Additionally, the concentrations of TauOs in the cerebrospinal fluid (CSF) and plasma of AD patients were higher than in non-demented controls, and revealed a negative correlation with mini-mental state examination (MMSE) scores. It was postulated that adding the measurements of TauOs to the panel of CSF biomarkers could improve the diagnosis of AD.

## 1. Introduction

### 1.1. Protein Misfolding

Protein folding is a physical process of the formation a highly organized molecular structure, with a characteristic, thermodynamically stable conformation, by a polypeptide chain [1]. Immediately after translation, the biosynthesis of proteins must be completed by post-translational processes. Proteins in the ‘native’ state form an unstructured polypeptide chain, which exhibits non-stable spatial structure. Their correct three-dimensional conformation is a result of the interactions between neighboring amino acids (AA). It is thought that AA sequence is the main determinant of the native structure of a given protein [2], although some observations indicate that the natural structure of proteins present in a living cell may also depend on interactions with other proteins or nucleic acids [1]. It should be emphasized that conformation of proteins is required for their physiological role and biological activity, whereas inappropriate spatial structure of given protein may lead to the formation of a molecule with altered biochemical and physical properties.

It was demonstrated that misfolding of proteins may be the result of various pathological processes, including mutations in the amino acid sequence, especially single-point dominant-negative mutations of the gene encoding the protein, its disturbed post-translational processing as well as trauma, ischemia, or oxidative stress [3]. Incorrect protein folding can also be due to a disruption of the normal folding process by external factors [4]. Single point mutations lead to the changes of conformation in the protein molecule and their polymerization. This mechanism is observed in sickle cell anemia [5], where β-globulin chains polymerize within the deoxygenated environment of tissue capillaries and cause the decrease in the elasticity of red blood cells [6]. Another example of diseases related to aberrant protein folding may be epidermolysis bullosa simplex, where the mutant forms of the keratin KRT5 and KRT14 antagonize the function of the wild-type protein, causing protein aggregation, resulting in harmful blistering of the skin in response to injury, particularly mechanical stress [7]. Incorrect protein folding occurs also in case of transthyretin (TTR), which normally is the primary carrier of thyroxine and a transporter of retinol (reviewed by [8]). While the active form of TTR is a tetramer, certain point mutations may destabilize its molecule, which results in the accumulation of TTR monomers and seeding amyloid formation. Misfolded forms of TTR include wild-type TTR (ATTRwt) or variant amyloidogenic TTR (ATTRv). The latter form is related to familial amyloid polyneuropathy (TTR-FAP), which is characterized by pain, muscular weakness and autonomic dysfunction. ATTRwt amyloidosis is considered as the cause of cardiomyopathy and, more recently, related to carpal tunnel syndrome. An introduction of disease-modifying therapies for this disease attracts many researchers [8]. A single-point mutation may be associated with favoring an intermediate form in the folding process as a more stable, the lowest energy state, over the properly folded protein. Therefore, stable protein aggregates can accumulate and disrupt cellular processes.

Protein misfolding and accumulation may occur within various tissues and cells. In neurons, the consequence of this accumulation are senile plaques, neurofibrillary tangles (NFTs), Pick bodies, Lewy bodies and other pathological changes in neuronal structure [9]. Amyloids may be the examples of such abnormal, misfolded proteinaceous structures. They are harmful, starch-like molecules organized in a supramolecular arrangement known as a cross-β structure, which arise from many different proteins and accumulate within various tissues [10]. Amyloids constitute of many copies of proteins folded into a shape allowing their molecules for sticking together. They form elongated fibrils consisting of β-sheet structures of separate monomers, which are positioned perpendicularly to the fibril axis and stacked strictly above each other. Moreover, identical polypeptides can fold into multiple distinct amyloid conformations. Interestingly, some amyloids may have physiological functions, such as formation of bacterial fimbriae, or hormone release and pigment deposition in human body [11].

The aggregation of proteins into amyloid fibrils and their subsequent deposition into plaques and intracellular inclusions is the characteristic feature of amyloid disease. Pathological amyloids are generated from normal proteins, when they lose their physiological functions and form fibrous deposits within tissues or intracellularly. The deposition of abnormal proteins can lead to a disruption of a normal function of organs. These changed proteins may even exert a toxic influence on cells and tissues, as it is seen in prion proteins [12]. The misfolding of proteins can trigger the further misfolding of other proteins and their accumulation into another aggregates or oligomers. It is known that certain amyloid proteins, called prions (for protein and infection), are known as contagious agents. The normal, 209 amino acids and 35 kDa form of the protein is called PrP^C^ and is found on the cell membranes. The physiological function of the prion protein remains poorly understood. It was found in animal models that the cleavage of PrP proteins in peripheral nerves causes the activation of myelin repair in Schwann cells, whereas the lack of PrP may cause their demyelination [13]. The infectious isoform of PrP, called PrP^Sc^, can act as a pattern to convert other non-infectious proteins into their infectious form [14]. Moreover, prion proteins are able to accumulate in stable aggregates within infected tissues, leading to their damage and cell death [15].

The accumulation and deposition of amyloid fibrils described as amyloidosis, is associated with many pathological conditions associated with ageing. What is more, amyloids may be the pathogenic forms of various proteins and polypeptides that can be linked with the development of more than 50 human diseases, known as amyloidoses including neurodegenerative diseases (NDs) [16,17]. Specifically, aggregated proteins are associated with prion-related disorders, such as bovine spongiform encephalopathy (BSE), also known as mad cow disease, or Creutzfeldt-Jakob disease (CJD). Furthermore, Alzheimer’s disease (AD), TTR-FAP and familial amyloid cardiomyopathy caused by hereditary transthyretin amyloidosis belong to amyloid-related illnesses, whereas Huntington’s (HD) and Parkinson’s disease (PD) are linked to intracellular aggregation of another misfolded proteins – huntingtin and alpha-synuclein, respectively.

### 1.2. Pathological Changes in Alzheimer’s Disease

For over than a century, since its description by Alois Alzheimer [18], the senile plaques within patients’ brain composed of amyloid beta (Aβ) and NFTs of Tau protein have been considered as the hallmark pathological features of AD [19]. The role of the accumulation of Aβ in AD has been widely discussed (for review see: [20]). It is believed that AD pathology is initiated by Aβ, especially the most amyloidogenic form Aβ_1–42_. It is also known that in AD this peptide may be present in various, antigenically distinct conformations, including monomeric and fibrillary forms, as well as oligomers of amyloid β peptide (AβOs) [21,22]. Currently, it is supposed that these soluble oligomers of Aβ, but not fibrillar Aβ42 accumulated within neuritic plaques, may be responsible for the toxic activity of Aβ, already on a very early stage of AD, perhaps even initiating pathological cascade. AβO toxicity includes loss of synapses, which is observed already in the earliest stages of AD [23], disruption of synaptic transmission, inhibition of long-term potentiation (LTP) and enhanced long-term synaptic depression (LTD) in the brain regions responsible for memory [24]. Moreover, is seems that intracellular soluble AβOs may contribute to the progressive nature of AD, spreading in a prion-like manner and can be transmitted between neurons, not only from one region to another within the brain, but it is even supposed that they might be transmitted between people [25].

Although convincing, the AβOs′ hypothesis has some limitations, that do not allow for explanation of known pathological processes in AD. One of them is that AβOs are not homogenous species. AβOs′ molecular weight, morphology and conformation are highly variegated, ranging from small, dimeric molecules about 4 kDa, through trimers, and low molecular weight (LMW) and high molecular weight (HMW) oligomers up to protofibrils and fibrils with molecular weight over 100 kDa. Such a phenotypic heterogeneity implies also a clinical and neuropathological diversity of AD, as well as variegated distribution of AβOs [25]. Moreover, binding of AβOs to cell membranes may be mediated by over twenty various proteins on the neuronal cell surface acting as AβO receptors (reviewed by [26]). Despite such big variety of putative receptors for oligomers, no single candidate receptor has been yet shown to be sufficient and essential for all features of AβO-mediated toxicity. This indicates that AβOs are ‘multifunctional’ and may have variegated toxic influence as drivers of neurodegeneration in AD. Therefore, it seems that AβOs hypothesis of AD etiology in not sufficient and does not explain all aspects of this disease. It may be confirmed by the fact that anti-Aβ therapies of AD, turned out to be not effective. As it was mentioned above, the progression of AD depends not only on AβOs, and may be complicated and accelerated by other pathologies, such as inflammation or Tau pathology.

Therefore, it was proposed that amyloid cascade is not the only pathway to AD (discussed by [27,28,29]). Although progressive deposition Tau protein in AD brain is preceded by the accumulation of Aβ, there is another hypothesis that assumes the abnormalities in the Tau protein are the first step in the cascade of pathological events in AD [30]. Additionally, this accumulation of Tau protein may contribute to pathological changes observed not only in AD, but also in other NDs, called tauopathies. Consequently, the aim of this review is to discuss the role of misfolded Tau protein in AD.

## 2. Materials and Methods

### Literature Search and Data Extraction

We have performed a comprehensive literature search using the PubMed electronic database from August 2018 to August 2019, with the following search strategy (key words sets): ‘Alzheimer’s disease AND Tau oligomers’ (612 items); ‘CSF AND Alzheimer’s disease AND Tau oligomers’ (31 studies) as well as ‘plasma OR serum AND Alzheimer’s disease AND Tau oligomers’ (31 papers). A total 674 records were found. Then we excluded 509 articles not containing data useful for our analysis. As a result, the remained 165 full-text articles and two books were assessed for eligibility. In the final step, we excluded 47 review papers, and so 113 original publications and 3 meta-analyses were included in the study. (Figure 1—PRISMA 2009 Flow Diagram).

## 3. Tau Protein

### 3.1. Tau Protein Gene and Its Transcription Products

Microtubule (MT) associated Tau protein is a heat stable protein of molecular weight approximately 60 kDa and 352–441 AA, which belongs to the family of microtubule-associated proteins (MAP) and is expressed mainly in neurons [32,33]. The gene *mapt* for Tau protein is located at locus 17q21 and includes at least 16 exons, which of them exons 0 and 14 are transcribed but not translated [34]. Two specific regions, N-terminal and C-terminal halves, may be distinguished in the primary structure of Tau protein, with N-terminal part showing higher variability among Tau proteins from different species [35]. These two major domains are divided on the basis of their microtubule interactions and AA character: C-terminal half is described as an ‘assembly domain’ whereas the N-terminal half is a ‘projection domain’. Furthermore, Tau comprises of four regions within its molecule: N-terminal region; proline-rich domain (PRD) which is located in the central region of this protein; region responsible for binding of Tau with MTs (microtubule binding domain, MBD), and C-terminal domain [36]. N-terminal region is encoded by exons 1–5, while exons 2 and 3 encode the 29 amino acids inserts in N-terminal region of Tau, which determine isoforms of this protein: 0N (no inserts), 1N (one insert encoded by exon 2), 2N (two inserts encoded both by exons 2 and 3). Exon 3 depends on the presence of exon 2, thus it is not transcribed in the absence of exon 2. Furthermore, the expression of exons 4A, 6 and 8 is limited to peripheral nervous system (PNS), encoding ‘big Tau’, a series of larger Tau proteins with molecular mass of 100–120 kDa [37]. This ‘big-Tau’ protein sequence is identical to the longest of small Tau isoforms, but includes an additional 254 AA insert in the N-terminal half of its molecule [37]. Exon 7 and the first half of exon 9 encode PRD, while MBD is encoded by exons 9–12. MBD includes four repeats, R1, R2, R3 and R4 (isoform 4R) or three repeats (R1, R3 and R4) (isoform 3R) of approximately 32 AA residues sequence. These residues include semi-conservative motif KXGS, where serine may be phosphorylated [38]. Number of these repeats depends on the presence of exon 10. Isoform 4R, including four repeats, is synthesized in the presence of this exon, whereas isoform 3R (three repeats) is produced when this exon is not expressed. The latter, C-terminal region is encoded by exon 13 [36,39] (Figure 2).

Moreover, there is a polymorphism of haplogroups, which differ in their orientation within the *mapt* gene: an inversion (H1) or non-inversion (H2) haplotype. Haplotype H2 is characteristic only for humans with European ancestry and is associated with increased expression of exon 3 in grey matter, what might be protective against some NDs [40]. On the contrary, inheritance of the H1/H1 genotype and the haplogroup H1 is linked to higher probability of certain NDs, such as AD, progressive supranuclear palsy (PSP), cortico-basal degeneration (CBD), idiopathic PD, and argyrophilic grain disease [41]. A greater risk of early dementia development in Down’s syndrome is related to a H1/H2 genotype [42].

The result of alternative splicing of exons 2, 3 and 10 is the presence of six Tau isoforms in human central nervous system (CNS) [43]. It is estimated that the ratio of isoforms containing 3R (not including exon 10) or 4R (including exon 10) is approximately 50/50 [44]. What is important, the proportion between isoforms 4R and 3R is essential for Tau protein function. Isoform 4R has higher affinity to MTs than 3R. Therefore, any change of 3R to 4R ratio may influence on Tau biological activity. It seems that alternative splicing of Tau, resulting in these two isoforms, may be also an essential factor for neurodegeneration [45]. Although both isoforms of Tau, 3R and 4R, may change the distribution of mitochondria within the cell body and reduce their localization to axons, they cause different alterations in retrograde and anterograde transport dynamics. It was demonstrated that 3R isoform has a slightly stronger effect on axon transport dynamics, what suggests that Tau-induced changes in axonal transport may underlie NDs [46].

It was revealed that the expression of Tau isoforms undergoes a specific ‘shift’ depending on the stage of development. In fetal period, only the shortest Tau isoform (0N3R) is expressed in the brain, whereas Tau maturation results in the presence of all the six isoforms of this protein in adult CNS [47]. These differences in the expression of Tau isoforms are associated with the formation of synapses, especially in sensory and motor cortices. Attenuated MT binding by fetal isoform 3R in the postnatal period, when compared to the 4R isoform, is critical for brain development and may be related to higher synaptic plasticity [44,48]. The developmental switch of splicing exon 10 of the Tau gene leads to the incorporation of an additional MT-binding repeat, resulting in increased stabilization of MTs [44]. Additionally, it seems that various populations of neurons may express different isoforms of Tau protein [36]. There are also regional differences in splicing of the *mapt* gene in the brain. For example, the amount of 0N3R tau in humans is lower in the cerebellum, while 4R Tau isoforms are increased in the globus pallidus than in other brain regions [38,49].

Furthermore, Tau molecule contains one or two cysteine residues: Cys322 located in R3 repeat, which is present in all isoforms, and Cys291 in R2 repeat, present only in 4R isoforms [50]. It seems that the balance between these isoforms may be important within the adult brain, and can influence on Tau aggregative properties in vitro [50]. It was shown that oxidation of Cys 322 in the repeat domain of Tau may control its assembly into larger aggregates, called paired helical filaments (PHFs) and NFTs. The redox potential in the neuron is crucial for PHF assembly, independently or in addition to pathological phosphorylation reactions.

### 3.2. Post-Translational Modifications of Tau

As it was mentioned above, each protein directly after translation, must undergo certain post-translational processes to complete its biosynthesis and gain biological activity. In case of Tau, these modifications include phosphorylation, glycosylation, ubiquitination, deamination, oxidation, and others. All post-translational changes may additionally complicate Tau isoforms [36]. Although various physiological processes regulate ability of binding Tau with MTs, the highest importance is ascribed to phosphorylation [45]. It may be confirmed by the fact that hyperphosphorylated Tau (hTau) is the main component of NFTs. The degree of Tau phosphorylation depends on the activity of certain enzymes, such as microtubule affinity regulating kinase (MARK), cycline-dependent kinase 5 (Cdk5), glycogen synthase kinase 3b (GSK3b) and protein kinase 2A [45]. These kinases regulate phosphorylation of Tau in the reaction of esterification of AA present in Tau molecule: serine (S), threonine (T) or tyrosine (Y).

Activity of Tau is closely related with its phosphorylation. There are up to 85 possible points of phosphorylation in the longest isoform of Tau in CNS (2N4R-Tau), which are located mainly in two flanking regions: the MBD and PRD domains [36,39]. The phosphorylation of these sites is significant for regulation of microtubule polymerization [51,52,53,54], controlling of axonal transport [55] and neurotransmitter receptors, such as AMPAR (α-amino-3-hydroxy-5-methyl-4-isoxazolepropionic acid receptor) and NMDAR (*N*-methyl-d-aspartate receptor) [56,57]. Moreover, overexpression of Tau protein inhibits kinesin-dependent trafficking of vesicles, mitochondria, and endoplasmic reticulum [58]. Phosphorylation of Tau regulates also its interaction with apolipoprotein E [59], Src-family kinases and neural plasma membrane [60,61,62] as well as interaction with DNA [63], and many others [64,65]. Relationships between Tau phosphorylation and its aggregation depend on the site where it occurs: certain residues are concerned with increased aggregation propensity [66], whereas in other sites it has the opposite effect, thus preventing aggregation [67].

Increased activity of kinases and decreased activity of phosphatases lead to an imbalance between these enzymes, resulting in hyperphosphorylation of Tau protein. Hyperphosphorylated Tau has lower affinity to MTs, induces their misfolding, destabilizes cytoskeleton and impairs axonal transport [36,39]. Interestingly, Tau hyperphosphorylation may also occur in hypothermia [68], starvation [69], chronic stress [70], or anesthesia [71,72].

### 3.3. Tau Protein Structure

Tau may be considered as an example of intrinsically disordered protein (IDP). IDPs are described as proteins without a fixed or ordered three-dimensional structure. They include a wide spectrum of states, from fully unstructured to partially structured molecules, which may comprise random coils, (pre-)molten globules, and large multi-domain proteins, that may be connected by various flexible linking domains [73].

Tau is a highly soluble protein, which is characterized by loose, open structure and natively unfolded conformation with less than 10% of α-helix and β-sheet regions [74]. However, aforementioned post-translational modifications of Tau, such as phosphorylation, acetylation, or its truncation, lead to alterations in the secondary structure that enable the appearance of α-helix or β-sheet regions in Tau molecule [75]. In solution, Tau molecule spontaneously tends to modify its conformation, favoring a paperclip-like shape [65,76]. In this conformation, the C-terminal end of the protein folds over into the proximity of the MDB repeats, but the N-terminus remains outside the reach of the repeat domain, although both ends of the molecule approach one another [76]. Such intramolecular interaction may prevent the self-aggregation of Tau by masking the regions involved in this process.

The paperclip-like Tau form differs from its extended conformation, which is required for interaction of Tau with MTs. It was demonstrated in the electron microscopic visualization that Tau binds to MTs in a linear alignment, lengthwise to the protofilament edges [77,78]. Each repeat of MBD interacts with one tubulin dimer and with its interface to the next tubulin [76,77,78,79,80,81]. When Tau is detached from MTs, they destabilize. Moreover, the post-translational modifications of Tau may also inhibit forming its paperclip-like structure, resulting in the increased aggregation propensity of this protein [82].

### 3.4. Biological Functions of Tau

The most recognized role of Tau as a scaffold protein, is the regulation of MTs stability, their assembly and spacing [83]. The adult neurons are the cells that not divide, so they need to stabilize their cytoskeleton in order to sustain neuronal processes such as the fast axonal transport through relatively long distances. Thus, normal Tau is an important factor in the MT-dependent axonal transport of signaling molecules and other substances [84]. Interaction of Tau with MTs occurs primarily through the repeated microtubule binding regions. In physiological conditions, Tau remains in a constant, dynamic balance with MTs, binding shortly with them, and after phosphorylation by kinases, a short-term disconnection from MTs occurs, but after the dephosphorylation by phosphatase, Tau is again re-attached to MT [85]. What is more, it seems that Tau is an essential factor for axonal elongation and maturation. It was demonstrated that Tau overexpression could induce the formation of neurites even in non-neuronal cells [86], while knockdown of *mapt* gene may inhibit neurite formation, as it was demonstraded in cultured rodent cerebellar neurons [87].

Whereas the activities of Tau presented above are located presynaptically and relate mainly with axonal part of neurons, the postsynaptic function of Tau remains unclear. It was demonstrated that this protein may be involved in the development of dendrites, as the important factor for neurite and axonal growth, and may contribute to synaptic plasticity [87]. These activities are associated with N-terminal region of Tau, which does not participate in Tau-MTs interactions. It was demonstrated that N-terminal region of Tau plays a role in neurite outgrowth in PC12 cells induced by nerve growth factor (NGF) [88]. PC12 cell line is derived from a pheochromocytoma of the rat adrenal medulla, and serve as cellular model for neurosecretion [89].

The novel roles of Tau, such as neuronal signaling pathways, DNA and RNA protection from oxidative stress, and regulation of synaptic functions, have been revealed. Targeting of Fyn, a tyrosine kinase from src family, in dendritic region of neurons is an example of postsynaptic Tau activity. The proline-rich sequence in the N-terminal half of Tau interacts with Fyn in cultured NIH3T3 cells. These cells function as the standard fibroblast cell line, whereas their co-transfection with Tau and Fyn kinase resulted in alterations in the morphology of NIH3T3 cells [88]. More recent studies demonstrate also a nuclear function of Tau, apart from its axonal and dendritic roles. It was revealed in neuronal cultures, that nuclear Tau may regulate transcriptional activity and maintain the integrity of DNA/RNA under hyperthermia, a known strong inducer of reactive oxygen species (ROS) [90]. This protein plays a protective role in neuronal DNA and RNA integrity in vivo, both under physiological conditions and in oxidative stress [91].

Another, novel role of Tau as a signaling molecule was also described in recent studies. It was revealed that this protein may regulate brain insulin pathway signaling [92]. Insulin signaling is known to control hippocampal plasticity and reference memory [93]. Insulin receptor substrate 1 (IRS-1) plays a key role in transmitting signals from the insulin and insulin-like growth factor-1 (IGF-1) receptors to intracellular pathways PI3K/Akt and Erk MAP kinase pathways. It was demonstrated on Tau knockout mice model that deletion of this protein leads to an impaired hippocampal response to insulin and promoted brain insulin resistance, caused by altered IRS-1 and PTEN (phosphatase and tensin homologue on chromosome 10) activities [92]. The clinical implications of these findings may be the fact that pathological loss of Tau function favors brain insulin resistance, what is one of the key factors for cognitive and metabolic impairments in AD patients [93].

### 3.5. Aggregation and Deposition of Misfolded Tau Protein

In addition to paperclip-like structure, Tau protein may also be present as a long, extended molecule, with ability of binding itself into Tau polymers or to other proteins [75]. Reactivity of monomers depends on the presence of thiol groups in cysteine residues. Free thiols in a reactive monomer allow formation of an intra- or intermolecular disulfide linkage [94]. Molecules of Tau with intramolecular disulfide linkage form a nonreactive monomer, whereas presence of one or more free thiols may result in reactive monomers, which are readily able to form an intermolecular disulfide connections and polymerization. Despite a low tendency of native Tau protein to accumulation [80], certain changes of its conformational status may induce the predisposition of this protein to assembly. It is known that Tau aggregation is a multistep process, which probably begins from the hyperphosphorylation of Tau and its detachment from MTs. In the process of aggregation, Tau is relocated to somato-dendritic regions of neurons, where further phosphorylation and structural changes of this protein occur (Table 1).

The result of self-aggregation of highly phosphorylated Tau may be soluble Tau oligomers (TauOs) of many different sizes, including dimers and trimers. It was revealed that any isoform of Tau may form Tau dimers. Two single Tau molecules interact through MBD domain with cysteine residues, forming a covalently linked Tau cysteine-dependent dimers, arranged in an antiparallel manner and linked by disulfide bonds [74]. Moreover, there are two distinct forms of dimeric Tau: reducible and cysteine-dependent dimers or non-reducible, cysteine-independent ones. The latter form has inter-molecular disulfide bridging at the MBD domain. Both dimeric Tau forms have been identified in vitro, as well as in transgenic mice JNPL3 expressing P301L mutation of Tau [97]. As it was mentioned before, 3R Tau isoform contains only one cysteine residue (Cys322 in R3 repeat) and may form its dimers only after oxidation. On the contrary, the 4R Tau isoform, having two cysteine residues (Cys291 in R2 and Cys322 in R3), can form various dimeric forms and an internal disulfide bridge [50]. These dimers are believed to be a crucial intermediate step in the formation of PHFs [50].

Although it is not fully elucidated if Tau dimers are the harmful form of this protein, the toxicity Tau trimers was revealed indeed [94]. It was demonstrated that trimeric forms resulting from two different Tau splice variants were noxious to human neuronal cells at low nanomolar concentrations, whereas monomers and dimers of this protein were not able to exert such effects [94]. Moreover, it was revealed that trimeric TauOs represent the minimal propagation unit, which may be spontaneously internalized by cells and seed further intracellular aggregation of this protein, mediating progression of Tau-related NDs [94]. Both recombinant Tau aggregates and Tau assemblies purified from AD brains with size three or more units were internalized by primary neurons and by HEK293 cells, which stably expresses the repeat domain of 2N4R Tau with two AD-associated mutations (P301L and V337M). Furthermore, it was shown that only 3-unit and larger assemblies from AD brain spontaneously seeded intracellular Tau aggregation in HEK293 cells [102]. These findings point out that the trimeric form is a minimum size of Tau assemblies that allows for spontaneous propagation of Tau aggregation from the outside to the inside of a cell [102].

The molecular size of small soluble Tau oligomers, both recombinant TauOs and isolated from human Tau transgenic mice, ranges between 120–180 kDa [97,103,104,105]. Moreover, small soluble Tau species, which probably included also Tau fragments, have been isolated from synapses in AD brains [106]. Their molecular weight was reported to correspond approximately dimer and trimer size. Apart from dimers and trimers, also small soluble TauOs containing six to eight Tau molecules and with approximately 300–500 kDa in size, have been described. It was demonstrated that these oligomers may develop from soluble dimeric Tau [97]. In addition, in mice overexpressing human Tau with the P301L mutation and NFTs, small Tau oligomers with a wide range of sizes were also detected [97].

Furthermore, Tau may aggregate into insoluble polymers, such as granular oligomers, straight filaments (SF), PHFs and NFTs. These granular aggregates, with globular shape are composed of approximately 40 monomers of Tau [98]. Their molecular mass reach approximately 1800 kDa, while diameter is only 20–50 nm, what indicates that granular TauOs are extremely densely packed [98]. This result in some difficulties in examination, separation and purification of these oligomers, which require certain changes in standard protocols for the fractionation of insoluble proteins. Although granular TauOs are fully insoluble, they are too small to sediment with a standard high-speed centrifuged insoluble fraction and to separate on a standard PAGE resolving gel [100]. It was assessed that sedimentation of granular TauOs requires a 200,000× *g* spin for 2 h [98].

It is not certain whether granular TauOs develop directly from small soluble oligomers in a linear pathway, or rather from monomers in the different pathway. Insoluble granular TauOs have β-sheet structure within their molecule and may be composed of partially folded monomeric Tau, what suggests that this type of TauOs may represent the intermediate step in the formation of fibrillar forms of Tau, such as PHFs or SFs [98]. It was shown that development of TauOs may be blocked by capping of cysteine residues with isoproterenol, an adrenergic receptor agonist including 1,2-dihydroxybenzene groups within its molecule [107]. Moreover, it was shown that isoproterenol may reverse certain emotional changes associated with the expression of P301L Tau and stimulate activity of neurons in the prelimbic frontal cortex [107].

It was revealed in vitro that Tau protein may aggregate further, forming fibrillar forms of Tau, such as SFs and PHFs [98]. SF strands are approximately 10 nm wide, therefore twisted fibrils of paired filaments, with approximately 80 nm periodicity, may exhibit alternating diameters 10 or 20 nm [100]. Recombinant smaller granules of TauOs could grow by binding with molecules of soluble Tau, whereas after long period of incubation, further binding to larger granular TauOs, exceeding a size of 20 nm, may result in the development of Tau fibrils [98]. Interestingly, extended time of incubation to 7 months resulted in the conversion of all Tau monomers and granular TauOs into filaments [98].

The time-dependent increase in Tau phosphorylation at specific sites and its abnormal molecule conformation lead to forming PHFs [43], which are highly insoluble and resistant to proteolysis pathological inclusions [101]. Normal Tau molecules include two phosphates [108], while Tau proteins isolated from AD brains is hyperphosphorylated and contains up to eight phosphates per molecule [109]. Therefore, the aberrations in Tau phosphorylation might be linked to its misfolding and deposition in the diseased brain [110]. Aggregated PHF are referred to as NFTs if they are formed within neuronal cell bodies, whereas aggregates formed in dendrites or axons are called neuropil threads [111].

Further PHFs aggregation leads to the development of NFTs and Pick bodies [39,45]. Although the final result of Tau accumulation are NFTs, the precise mechanism of their formation and the sequence in which these events occur is not clearly elucidated. It is neither fully understood whether Tau aggregates represent a primary causative factor in AD or they play a more peripheral role. It was shown that increased ability of Tau for forming PHFs and NFTs is related to disturbed processes of phosphorylation of flanking regions of Tau at certain AA, such as serine 396 and serine 404 [112]. Moreover, Tau phosphorylation at Ser396 and Ser404 by human recombinant Tau protein kinase II inhibited Tau’s ability to promote assembly of MTs [113]. It was also demonstrated that the phosphorylation at these sites is one of the earliest events in AD and in Down syndrome [114].

NFTs, first described by Alzheimer in 1907, belong to the pathological hallmarks of AD, although they may also occur in other tauopathies [75]. In AD, NFTs are composed of Tau molecules accumulated to fibrillar aggregates, PHFs and SFs, which may fill the entire neuronal cytoplasm. PHFs differ from SFs the presence heparan and chondroitin sulfates, which are associated to Tau polymers [115]. The process of Tau polymerization into fibrils may be facilitated by various polyanionic cofactors, including glycosaminoglycans like heparin and neuroparin [116], sulphoglycosaminoglycans such as keratins or chondroitin sulfates, fatty acids such as arachadonic acid as well as alkyl sulfate detergents [100]. Moreover, aggregation of Tau and formation of fibrils may be accelerated by metal ions (Pb^2+^, Zn^2+^, Cd^2+^, Cu^2+^, Mg^2+^) [117,118]. Furthermore, it was shown that Tau filaments may spontaneously cluster together into NFTs in vitro [100].

The formation of NFTs may be affected already on the early stages of this process. Although the search for mutations associated with *mapt* gene, that could be related with AD, have failed, several factors that affect the NFTs development have been identified so far [112]. They include some genetic factors [119,120], exposure to toxins such as nicotine [121] or aluminum [122,123] as well as neurotoxicity of Aβ [124]. Moreover, various pathological processes, such as inflammation [125,126] and oxidative stress [127] may also have an impact on generation of NFTs. Additionally, impaired cholesterol metabolism and atherosclerosis [128,129,130], dietary deficits [131,132], depression [133,134] and stress [135] may also contribute to formation of NFTs.

The localization of NFTs within AD brain changes depending on the progression of AD, and correlates with the severity of the cognitive decline. Therefore the topographic staging of NFTs may be applied for the pathological diagnosis of AD [136]. The Braak stages define the degree of spatiotemporal NFT’s involvement in AD: changes limited to brain transentorhinal region are described as the Braak stages I and II, the involvement of limbic regions such as the hippocampus represents stages III and IV, whereas stages V and VI correspond to extensive neocortical NFTs burden [136]. Moreover, the clinical manifestation of AD and the degree of cognitive impairment is often better reflected by NTF counts than by burden of Aβ plaques in brain [137].

Fibrillar forms of Tau, PHFs and SFs are not only the components of NFTs, but they may form neuropil threads, which are aggregates of hTau in dystrophic neurites largely displacing the cytoskeleton [103]. Neuropil threads are rather early pathological inclusions, which may be also accompanied by Tau dimers and oligomers. These neuropil threads are also the morphological hallmarks of AD [138]. It has been suggested that the threads may play a role in the cognitive impairment seen in AD [139]. Primarily, NFTs are intraneuronal aggregates of misfolded hTau, but after the death of tangle-bearing neurons, they became the extraneuronal structures called ‘ghost’ tangles. Ghost tangles are the remnants of degenerated neurons within which NTFs have been formed. They are composed of Tau filaments and fragments, which have undergone substantial proteolysis. Tau molecules within ghost tangles are loosely arranged in large extracellular bundles [100].

## 4. Toxic Activity of Tau and TauOs in AD

### 4.1. Tau Protein Toxicity Related to Its Post-Translational Modifications and Missorting

Although mechanisms of toxic activity of Tau are not fully recognized, it is supposed that the Tau toxicity is related rather not to insoluble Tau aggregates, but to its intermediate forms. It seems that NFTs themselves, despite being composed of toxic Tau, are probably neither necessary nor sufficient for Tau-induced neuronal dysfunction and toxicity. For example, in some animal models of tauopathy, notwithstanding evident neurodegeneration, tangles are not formed at all, or rather later, after the occurrence of cognitive/behavioral impairments and neuronal death [95,140,141,142]. It was proposed that formation of NFTs might be somehow a failed kind of neuronal protective response against toxic Tau seeding and a form of its toxic aggregates sequestration [45].

Whereas physiological Tau is soluble and non-toxic, under pathological conditions this protein may undergo some modifications and achieve toxic features, even though it may remain soluble. One of the results of post-translational modifications may be the detachment of Tau from microtubules, leading to disassembly of MTs in axons [95]. When disconnected from MTs, Tau diffuse rapidly into other neuronal compartments and, by aberrant sorting within neuronal cell, it may induce synaptic dysfunction. Furthermore, entering of pathological forms of Tau into postsynaptic compartments and dendrites causes a decrease in the number of synaptic vesicles and, finally, loss of the synapses [96]. However, it was revealed in more recent study that missorted Tau in dendrites is not derived from axons but it originates from Tau newly synthesized in the cell body [111].

### 4.2. Relationship between Toxic AβOs and TauOs

Postsynaptic dysfunction of Tau is also associated with a reduction of dendritic spines and local increase in Ca^2+^. It was observed that this missorting of Tau into dendrites may be caused by AβOs exposure of primary hippocampal neurons [96]. It was also revealed that Tau knockout neurons are resistant to AβO toxicity, whereas introduction of Tau could re-sensitize these cells to Aβ-induced spine loss and disintegration of MTs [111]. Activity-dependent Tau translocation to excitatory synapse may be disrupted by exposure to AβOs [143]. Interestingly, this AβOs-induced somatodendritic missorting occurs not only with Tau, but also with other axonal proteins such as neurofilaments [142]. Moreover, hyper-phosphorylation of Tau and NFT formation, may be induced by AβOs either [144].

It was also demonstrated that AβOs toxicity may be mediated by the dendritic function of Tau [144]. The NMDAR receptor, which is believed as AβOs receptor, is the substrate of Fyn kinase. It was suggested that the tyrosine phosphorylation of Tau by Fyn kinase may also have a role in neuropathogenesis of AD, where upregulation of Fyn is observed [88]. Both in transgenic mice expressing truncated form of Tau (ΔTau), which does not include MBD, or in Tau (−/−) mice, the postsynaptic targeting of Fyn was disrupted. The disturbed targeting of this enzyme resulted in alleviation of the excitotoxicity mediated by NMDAR and, finally, the toxicity of AβOs [144]. These results demonstrate that both expression of truncated Tau and the deficiency of this protein may prevent memory deficits and improve survival in a mice model of AD. Moreover, the expression of ΔTau or deficiency of normal Tau prevented memory deficits in APP23 mice, which are Aβ forming animal model of AD, and improved their survival [144]. Additionally, these deficits were fully rescued in vivo with a peptide uncoupling the Fyn-mediated interaction of NMDAR and another postsynaptic protein PSD-95 (post synaptic density 95). These results suggest that this Tau dendritic activity may result in postsynaptic Aβ toxicity. The associations of Tau with synaptic proteins pave the way to possible direct implications for the pathogenesis and treatment of AD and indicate that dendritic activity of Tau may promote postsynaptic toxicity of Aβ. These findings indicate that there is a link in AD pathology between oligomeric forms of various proteins, including amyloid β and Tau.

### 4.3. Influence of Tau on Synaptic Transmission

It was shown that inhibition of Tau expression may rescue memory deficits in mice expressing mutated form of Tau_P301L_, a mouse model of neurodegeneration [145]. After the suppression of transgenic Tau by doxycycline administration, the recovery of memory function and stabilization of neuron numbers were observed. However, suppression of Tau did not induce change in NFTs burden. It seems that tangles themselves are insufficient to cause the cognitive decline or neuronal death in this model of tauopathy [145]. On the contrary, it was suggested that NFTs represent rather a stable, favorable state that probably acts as a sink for remaining Tau_P301L_species. Furthermore, it appears that in AD this degenerating sequence of events, although triggered by Tau overexpression, may branch from a Tau filament formation pathway. Tau may produce aggregates with toxic properties already before forming insoluble fibrils.

It was demonstrated on another animal model of tauopathy that mice expressing highly pro-aggregant ΔK280 mutation of this protein can develop synapse loss and various pathological features of Tau, including missorting, phosphorylation and early pretangle formation [146]. Moreover, hippocampal Tau pathology in pro-aggregant mice correlated with impaired synaptic plasticity, loss of LTP in hippocampus and a significant reduction of synaptic proteins and dendritic spines as well as with severe cognitive deficits in learning/memory performance tests. Interestingly, memory deficits and LTP depression recovered, when pro-aggregant Tau was turned off. Moreover, switching-off the pro-aggregant Tau resulted in reduction of its phosphorylation and reversal of Tau missorting, although the insoluble forms of Tau were still present. These findings demonstrate that Tau-induced impairments might be reversible after turn off pro-aggregant Tau [146].

### 4.4. Toxic Tau Aggregates

Growing body of evidence suggests that the soluble species of TauOs, observed already at early stages of AD pathology, seem to be the most deleterious forms of Tau. The prefibrillar TauOs, similarly to AβOs, are characterized by the neurotoxicity, infectivity, and ability for amplification, which are the mechanisms that underlie progression of AD, leading to the death of neurons [103,147,148,149].

It was revealed that pre-fibrillar TauOs may also alter the protective function of normal Tau for nucleic acids in hippocampal neurons in vivo [150]. Pathological Tau cannot enter the nucleus, which may result in DNA damage owing to the loss of the DNA-protective function. It was demonstrated in a mice model of tauopathy, that hyperthermia may selectively induce the oxidative damage and trigger the strand breaks of nucleic acid in hippocampal neurons that display early Tau phosphorylation [150]. It was also revealed that the loss of nuclear Tau protein affects the structure, transcription and repair of neuronal peri-centromeric heterochromatin [151].

Another toxic feature of pathological Tau is the influence on neuronal activity and induction of their hyperexcitability. Besides cognitive impairment, AD has been also referred to as a ‘disease of synaptic failure’, and may be associated with an increased incidence of unprovoked epileptic-type seizures [152]. Epileptiform activity in AD may be interpreted as a secondary process resulting from advanced stages of neurodegeneration [153]. It was demonstrated that antisense reduction of Tau could suppress epileptic seizures in animal models of AD, what indicates that Tau may be involved in the regulating neuronal hyperexcitability [154]. Tau ablation reduced the frequency of seizures in mice model of epilepsy and prevented the high mortality in these animals [155]. Moreover, it prevented biochemical changes in the hippocampus symptomatic for epileptic activity and rescued defects in learning and memory in these mice.

It was shown that normal Tau is essential for long-term depression in the hippocampus [156]. However, by forming toxic aggregates, Tau severely disturbs the neuronal function, especially synaptic transmission. The number of ultra-stable TauOs visualized in brightfield microscopy was estimated to accumulate within trillions of synapses in AD brain cortex, what outnumbers macroscopic Tau aggregates such as NFTs by 10,000-fold [157]. The toxic influence of Tau on synapses may be also related with dissemination of misfolded forms of this protein, especially TauOs. Similarly to AβOs and prion proteins, AD brain-derived TauOs may propagate pathology from endogenous Tau, spreading within the living brain and making AD an infectious disease [158]. This distribution may occur when TauOs, released by neurons into the extracellular space, are taken up again by other neurons [157]. An ability of misfolded Tau to transmit across synapses was demonstrated both in AD animal models [159] and human cortical neurons from AD brains [157]. The immunostaining against Alz-50, an antibody which recognizes misfolded Tau in AD-affected brains, was strongly elevated in synaptoneurosomes isolated from AD than in non-demented control brains. This result indicates that Tau may undergo hyperphosphorylation, misfolding, and oligomerization at AD-affected synapses [157]. It was hypothesized that the infectious TauO species might be transmitted through synapses by synaptic vesicular trafficking [160]. Frequent and symmetric deposition of misfolded TauOs within presynaptic and postsynaptic terminals observed in AD suggests that this vesicular transport of oligomers is possible in either direction of transmission [157]. Another proposed explanation is the uptake of TauOs in neighboring synaptic terminals by accidental endocytosis. Incompletely degraded TauOs, which may remain as debris in the extracellular space, also allows for seeding further Tau misfolding [157].

Since it have become that insoluble aggregates of Tau, such as NFT or PHS might not be toxic, an assumption that granular TauOs perhaps might be the most toxic species of Tau emerged. Granular TauOs are the type of Tau aggregates, which have been isolated from AD brains [99]. These TauO fractions derived from human frontal cortex were detected not only in the samples from AD patients with advanced stages of the disease, characterized by large amount of NFTs, but even in samples from patients with no NFTs in the frontal cortex [98]. Additionally, granular oligomers isolated from the brains of patients with early stages of AD were smaller than those from advanced AD. These results suggest that granular TauOs accumulate before forming of Tau fibrils. Interestingly, it was demonstrated that granular TauOs appear also in non-AD brains, whereas no histologically verified NFTs were observed in these patients [98]. What is important, an increase in granular TauO levels occurs before NFTs form and before manifestation of clinical symptoms of AD, suggesting that elevation in TauOs level may represent a very early sign of brain aging and AD [99]. The summary of physiological functions and possible toxic activities of Tau is presented in Table 2.

## 5. Diagnostic Possibilities of TauOs Determination

There is a lack of single laboratory biomarker for the diagnosis of AD. The recommendations for diagnosing AD comprise the evaluation of Mini Mental State Examination test (MMSE), as well as neuroimaging techniques, such as magnetic resonance imaging (MRI) and positron emission tomography (PET) scans of the brain. The concentrations of Aβ42 and Tau proteins (total Tau and its phosphorylated form Tau_181_) in the cerebrospinal fluid (CSF) are considered the major CSF biomarkers in AD pathogenesis, and they have been included as research criteria for the diagnosis of AD since 2007 [161]. However, it should be highlighted that the current clinical criteria of AD [162] and MCI [163] do not include the use of laboratory biomarkers in everyday routine. Apart from these core CSF biomarkers, neurofilament light protein (NFL) and plasma total Tau have also been reported to be associated with AD [164].

Taking into consideration the role of TauOs in pathogenesis of AD, it could be helpful if the detection and measurement of their CSF and/or blood levels was also possible. Since accumulation of prefibrillar pathological forms of Tau, especially TauOs, is rather an early event in the progression of AD pathology, it is reasonable to develop laboratory tests for their measurement, probably with addition to assays for AβOs as the biomarkers which could facilitate pre-symptomatic diagnosis and staging of AD.

Therefore, there is a need to generate the reagents, especially specific antibodies, which could selectively recognize oligomeric forms of Tau. There were isolated three different single chain antibody fragments (scFvs): F9T, D11C and H2A, which may recognize naturally occurring Tau aggregates either produced by cultured neurons or naturally present in brain tissue from an AD mouse model or human AD brain samples [165]. It was demonstrated that these scFvs specifically bind toxic TauOs, but not monomeric or fibrillar tau. Moreover, they could detect the presence of oligomeric Tau in mouse brain samples from mice model of AD earlier than NFTs may be typically identified. In addition, these antibody fragments could also differentiate between AD and cognitively normal post-mortem human brains, what demonstrates the possibility of developing the biomarkers for early detection and progression of AD [165].

More recent studies demonstrated the presence of TauOs in samples of CSF collected from patients with various degrees of cognitive impairment, which was confirmed in Western blot analysis with anti-TauOs antibodies, T22 (rabbit polyclonal), and TOMA (mouse monoclonal) [166]. Concentrations of TauOs were elevated in AD patients compared to age-matched controls. Similarly, the ratio of oligomeric to total Tau (TauO/t-Tau) was increased in AD patients, with the higher values in moderate to severe AD group than in mild AD and non-demented controls [166]. These results suggest that the measurements of TauOs in CSF could be added to the panel of CSF biomarkers for the diagnosis of AD.

Although Tau is an intracellular protein with certain functions inside the cell, it may be either secreted actively into the extracellular space, what could be a physiological response of neurons to increased intracellular amounts of this protein during the progression of AD. Furthermore, the functions of the blood-brain and blood-CSF barriers may be disturbed in AD, where AβOs could damage their integrity by activation of matrix metalloproteinases (MMPs) [167]. Thus, in AD AβOs and MMPs may additionally contribute to the passage of Tau and other proteins from CSF to the blood [167].

The diagnostic utility of TauOs as early plasma biomarker of AD was assessed using D11C, another scFvs that selectively binds oligomeric Tau [168]. Longitudinal analysis of plasma samples demonstrated that levels of D11C-reactive TauOs were significantly higher in AD patients with dementia than in non-demented controls and these levels increased with the progression of AD [168]. Concentrations of TauOs in the sera of AD patients and aged controls were also measured using anti-oligomeric antibody T22 [169]. Although TauOs were present in the serum of AD patients, they could be detected also in non-demented subjects and correlated positively with age. These findings suggest that clearance of extracellular Tau proteins might take place in the periphery, outside the CNS. What is important, in patients with mild cognitive impairment (MCI) and MCI due to AD (MCI-AD) serum levels of TauOs were lower than in control group. It seems that the decreased levels of TauOs in MCI patients, could be the result of impaired clearance of Tau from brain to blood and consequent extracellular accumulation of Tau aggregates.

## 6. Conclusions

In the current study we have examined literature on the role of various species of Tau in synaptic dysfunction, memory loss, as well as in seeding and spreading of AD. Although the causative role of Aβ cascade in the development of AD has been the main explanation of pathological processes of this devastating disease for last 25 years, it became evident, that this hypothesis is not sufficient for elucidation all its aspects. Amyloid β theory has been recently reconsidered by pointing attention to oligomeric forms of Aβ, which probably are the toxic species of amyloid. However, this complemented hypothesis also does not explain all pathological features of AD. Additionally, since Aβ plaques and NFTs co-occur in AD with different topological and temporal patterns, the scientific attention has shifted to another protein with possible influence on the development of AD, Tau protein, which is the main component of these aggregates.

It was demonstrated that in AD the pathological role may be assigned to Tau, especially its aggregated forms. It was shown that NFTs better than Aβ plaques correspond with the duration and severity of AD, predicting cognitive status in this disease. Moreover, it was suggested that rather soluble oligomers of Tau than insoluble aggregates of this protein may be responsible for harmful effects in AD. It is difficult to precisely determine the most toxic form of Tau, which may be present in multiple, various soluble species, possessing very different properties, as well as macroscopic insoluble fibrils. Besides the six splice variants of Tau (0N, 1N, 2N multiplied by 3R and 4R), there are three major conformations described (paperclip-like, nonreactive with internal disulfide bridge and reactive with free thiol groups). This abundance of isoforms is complicated by the array of various phosphorylation and other post-translational combinations. Furthermore, two conformations of dimeric Tau depending on cysteine, as well as Tau trimers, and various sizes and phosphorylation states of TauOs were described. All these species of Tau may have importance in the development of AD, but it seems that the soluble oligomers such as granular TauOs are the most detrimental form of Tau. On the contrary, formation of insoluble NFTs was suggested rather as a protective response of damaged neurons than as a necessary precursor of the neuronal death in AD.

There is an urgent need to develop simple, reliable and sensitive biomarkers of AD that can both diagnose AD at an early enough stage, when therapies can be effective, and be used for screening the treatment of AD. Ideally, these biomarkers would be blood-based, since collecting blood samples is an easier and less invasive intervention than lumbar puncture. Is seems reasonable that TauOs could be promising candidates for such biomarkers of AD, especially TauOs in plasma or serum of AD patients. However, additional studies are needed before implementation of the analysis of TauOs as a diagnostically useful method for the routine clinical assessment of AD patients. The development of validated biomarker tests by measuring TauOs in the blood would require performing the series of robust, well standardized assays with appropriate high sensitivity. Before implementation, these assays should also be confirmed on large cohorts of patients in critical multi-centre comparison.

## Figures and Tables

**Figure 1 ijms-20-04661-f001:**
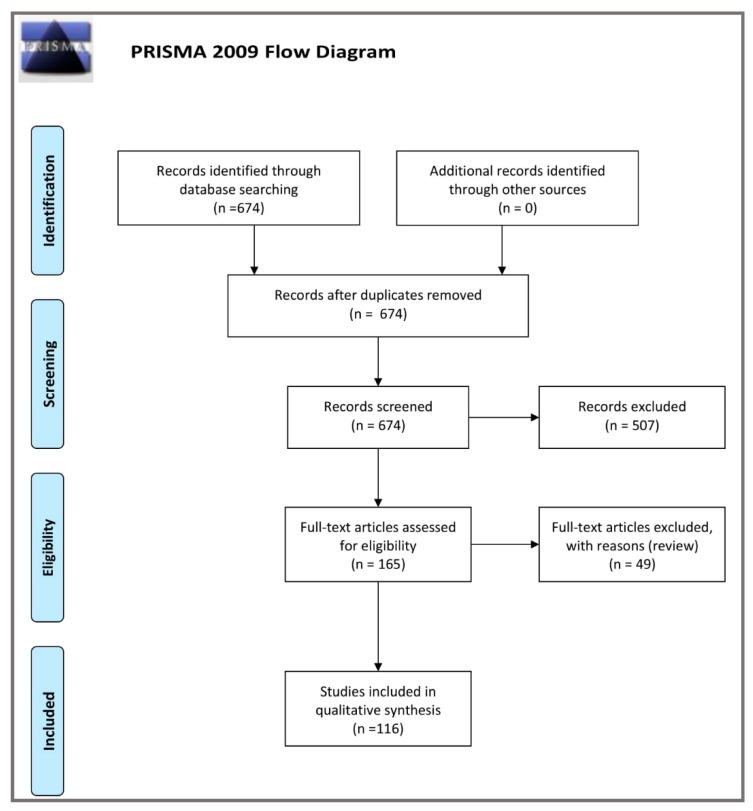
Schematic illustration of articles included in the review manuscript [31].

**Figure 2 ijms-20-04661-f002:**
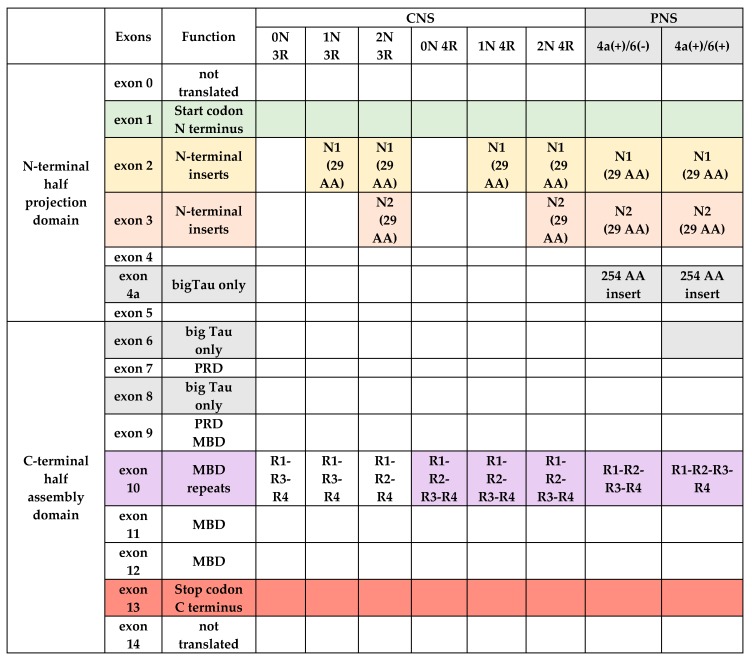
Schematic structure of Tau molecule in central (CNS) and peripheral nervous system (PNS) in relation to alternative splicing of exons. Abbreviations; PRD—proline rich domain, MBD—microtubule binding domain, AA—aminoacids.

**Table 1 ijms-20-04661-t001:** Major forms of Tau identified.

Species of Tau	Molecular Weight and Size	Toxicity
Monomer	60 kDa, 352–441 AA	no
Abnormally phosphorylated non-PHF Tau (AD P-Tau)	67–70 kDa	yes
		detachment of P-Tau from MTs leads to their disassembly in axons [95] missorting of P-Tau: induction of synaptic dysfunction decrease in the number of synaptic vesicles loss of the synapses [96]
Dimer/trimer	120–180 kDa, length 22–25 nm	yes, some types trimeric TauOs:
		toxic for neurons at nanomolar levels [94] minimal propagation unit [94]
Small soluble oligomer (6–8 molecules) (TauOs)	300-500 kDa	yes, some types
		TauOs containing 6–8 molecules: may develop from soluble dimeric Tau [97] detected in mice model of tauopathy [97]
Granular tau oligomers (36 Tau monomers) (gTauOs)	1800 kDa, diameter 20–50 nm	yes, some types
		intermediate step in the formation of PHFs and SFs [98] isolated from AD brains [99]
Straight filaments (SF)	>50 nm length, 10 nm width	not always
		Tau hyperphosphorylation may induce its self-assembly into NFTs consisted of PHFs/SFs [100]
Paired helical filaments (PHF)	10–20 nm width, with 80 nm periodicity, length > 220 nm	probably not toxic
		highly insoluble and resistant to proteolysis [101]
Neurofibrillary tangles (NFT)	NA	probably not toxic
		form of the sequestration of toxic aggregates in neuronal protective response [45]
Ghost tangle	NA	probably not toxic
		remnants of degenerated neurons within which NTFs have been formed [100]

NA—not assessed.

**Table 2 ijms-20-04661-t002:** Functions of normal Tau and toxic activity of pathological forms of Tau.

Physiological Functions of Tau	Pathological Activity of Tau
Main distribution in normal healthy neurons in axons [82,83]: Scaffold protein Stabilization and assembly of MTs Regulation of axonal transport Activity through the repeated regions of MBD	Results of post-translational modifications [95,96]: Detachment of tau from microtubules Microtubule disassembly in axons Mislocalization in presynaptic terminals Induction of synaptic dysfunction Reduction in the number of synaptic vesicles in presynaptic terminals Synapse loss
Dendritic function associated with N-terminal region of Tau [87]: Detected in dendrites in small amounts Involved in the development of dendrites Important factor for neurite and axonal growth Contribution to synaptic plasticity Neurite outgrowth	Missorting of pathological Tau into dendrites and postsynaptic compartments [96,157]: Induction of postsynaptic dysfunction mediated by Aβ oligomers Synapse loss Dendritic pathological tau [144]: Transport of FYN kinase to the postsynaptic sites. Reduction of dendritic spines and local increase in Ca^2+^. Enhancement of Aβ toxicity
Nuclear function [89,90]: DNA and RNA protection from oxidative stress Regulation of transcriptional activity Maintaining the integrity of genomic DNA and RNA in hyperthermia	Loss of possibility to enter the nucleus by pathologic Tau [93]: Loss of the DNA-protective function of tau DNA damage
Neuronal signaling pathways—PRD domain [91]: Regulation of brain insulin pathway signaling	Signaling molecule in the postsynaptic compartment [144,146]: Influence on neuronal activity Disturbed synaptic transmission Loss of LTP in hippocampus Impaired synaptic plasticity Significant reduction of synaptic proteins and dendritic spines Strengthening the excitotoxicity of AβOs by excitatory neurotransmitters NMDA and PSD95 Induction of neuronal hyperexcitability Cognitive deficits in learning/memory performance tests in mice model of tauopathy
	Formation of aggregates [99,102,112]: Disturbed neuronal function Release of Tau aggregates into extracellular space Uptake of Tau aggregates by other neurons Spread of Tau pathology
	Tau knockout [93]: Impaired hippocampal response to insulin and promotion of brain insulin resistance

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
