# Peer review of "The Role of Protein Misfolding and Tau Oligomers (TauOs) in Alzheimer′s Disease (AD)"

_ijms, 2019, doi:10.3390/ijms20194661_

Round 1

Reviewer 1 Report

In this review, the authors summarize the role of misfolded tau protein, which are deeply related to pathogenesis of Alzheimer’s disease and other neurodegenerative diseases. Although the detailed mechanisms of misfolded tau protein acting on Alzheimer’s disease are still unclear, they tried to describe the various roles and effects of the toxicity soluble oligomers of tau protein. This review is informative and is means to publish in International Journal of Molecular Sciences with minor correction.

Figure 1, Figure 2 and Table 2 in page 3 line 139, page 4 line 166, and page 10 line 440 are missing in the manuscript. Table 1 is informative, but if the authors add the detailed experimental conditions of tau protein with the references in the table, it is more valuable. To improve the readability, it is necessary to divide the section 4 with subheadings.

Author Response

Response to Reviewer 1 Comments:

In this review, the authors summarize the role of misfolded tau protein, which are deeply related to pathogenesis of Alzheimer’s disease and other neurodegenerative diseases. Although the detailed mechanisms of misfolded tau protein acting on Alzheimer’s disease are still unclear, they tried to describe the various roles and effects of the toxicity soluble oligomers of tau protein. This review is informative and is means to publish in International Journal of Molecular Sciences with minor correction.

Thank you very much for this general positive comment.

Figure 1, Figure 2 and Table 2 in page 3 line 139, page 4 line 166, and page 10 line 440 are missing in the manuscript.

Thank you very much for this remark. These Figures and Tables have been inserted in appropriate places in the new version of the manuscript (pages 4, 5, and 14-15, respectively).

Table 1 is informative, but if the authors add the detailed experimental conditions of tau protein with the references in the table, it is more valuable.

Thank you for this valuable suggestion. The experimental conditions of Tau protein with the references have been added to the Table 1 in the new version of the manuscript (Page 9).

To improve the readability, it is necessary to divide the section 4 with subheadings. 

Many thanks for your valuable suggestions. These subheadings have been added in the revised version of the paper (pages 12-14).

Reviewer 2 Report

The authors thoroughly reviewed articles regarding the role of misfolded Tau protein in Alzheimer’s disease, particularly focusing on small soluble oligomers. Wide-ranged issues from molecular aspects to diagnosis issues are described.

This is an interesting article exploring the toxicity and diagnostic values of small soluble Tau oligomers. It will attract broad range of readers from basic researchers to physicians. The manuscript is well written, and I enjoyed reading it.

Although I do not have any critical comments, minor issues to strengthen this manuscript are raised as follows: 

On page 2, the authors enumerate sickle cell anemia and epidermolysis bullosa simplex as diseases caused by protein conformational change associated with single point mutation. I would suggest adding hereditary transthyretin amyloidosis because an introduction of disease-modifying therapies for this disease attracts many researchers (Biomedicines 2019; 7: E11). “polyneuropathy and familial amyloid cardiomyopathy” in lines 87 to 88 on page 2 would be hereditary transthyretin amyloidosis “. Please refer to a recent nomenclature recommendation (Amyloid 2018; 25: 215-219).

Author Response

Response to Reviewer 2 Comments

The authors thoroughly reviewed articles regarding the role of misfolded Tau protein in Alzheimer’s disease, particularly focusing on small soluble oligomers. Wide-ranged issues from molecular aspects to diagnosis issues are described. This is an interesting article exploring the toxicity and diagnostic values of small soluble Tau oligomers. It will attract broad range of readers from basic researchers to physicians. The manuscript is well written, and I enjoyed reading it.

Thank you very much for your kind general comment!

Although I do not have any critical comments, minor issues to strengthen this manuscript are raised as follows: On page 2, the authors enumerate sickle cell anemia and epidermolysis bullosa simplex as diseases caused by protein conformational change associated with single point mutation. I would suggest adding hereditary transthyretin amyloidosis because an introduction of disease-modifying therapies for this disease attracts many researchers (Biomedicines 2019; 7: E11). “polyneuropathy and familial amyloid cardiomyopathy” in lines 87 to 88 on page 2 would be hereditary transthyretin amyloidosis“.

Thank you very much for this remark. This issue has been discussed in the new version of our manuscript (pages 2-3).

Please refer to a recent nomenclature recommendation (Amyloid 2018; 25: 215-219).

Thank you very much again for this comment. We carefully checked the whole manuscript

according to recent nomenclature recommendations, as it was suggested.